# An Evaluation of the Health Economics of Postnatal Depression Prevention and Treatment Strategies in China: A Cost-Effectiveness Analysis

**DOI:** 10.3390/healthcare12111076

**Published:** 2024-05-24

**Authors:** Zhou Zheng, Tianyu Feng, Jiaying Xu, Xiaolin Zhang, Xihe Yu

**Affiliations:** School of Public Health, Jilin University, Changchun 130022, China; zhengzhou19@mails.jlu.edu.cn (Z.Z.); fengty21@mails.jlu.edu.cn (T.F.); xjy21@mails.jlu.edu.cn (J.X.); zhangxl21@mails.jlu.edu.cn (X.Z.)

**Keywords:** cost-effectiveness analysis, postpartum depression, women’s health, mental healthcare

## Abstract

Objective: The primary objectives of this study are to assess the cost-effectiveness of early postnatal screening and prenatal psychological interventions for the prevention and treatment of postpartum depression (PPD) among Chinese pregnant women. Additionally, we aim to explore the most cost-effective prevention and treatment strategies for PPD in China. Methods: We used TreeAge 2019 to construct a decision tree model, with the model assuming a simulated queue size of 10,000 people. The model employed Monte Carlo simulation to assess the cost-effectiveness of PPD prevention and treatment strategies. Transfer probabilities were derived from published studies and meta-analyses. Cost and effectiveness data were obtained from published sources and relevant studies. Incremental cost-effectiveness ratios (ICERs) were used to describe the results, with willingness-to-pay (WTP) thresholds set at China’s gross domestic product (GDP) per capita. Results: Compared to the usual care group, the cost per additional quality-adjusted life year (QALY) for the early postnatal screening group and the prenatal psychological interventions is USD 6840.28 and USD 3720.74, respectively. The cure rate of mixed treatments for PPD has the greatest impact on the model, while patient participation in treatment has a minor impact on the cost-effectiveness of prevention and treatment strategies. Conclusion: Both early postnatal screening and prenatal psychological interventions are found to be highly cost-effective strategies for preventing and treating PPD in China. Prenatal psychological interventions for pregnant women are the most cost-effective prevention and treatment strategy. As such, from the perspective of national payers, we recommend that maternal screening for PPD be implemented in China to identify high-risk groups early on and to facilitate effective intervention.

## 1. Introduction

Postpartum depression (PPD) is a prevalent psychiatric disorder among women during the perinatal period [1], affecting nearly 12% of women globally [2,3]. In China alone, between 10 and 16% of pregnant women experience PPD [4]. As a result, PPD has become a significant public health concern [5]. Often occurring within the first six weeks after delivery, PPD can have numerous adverse health consequences for both mother and child, including chronic maternal depression and harm to the infant [6,7]. Furthermore, PPD can strain relationships within the family unit, placing a heavy burden on all involved [8].

PPD is a devastating and costly illness that is the most serious mental health concern among pregnant women. Not only does PPD impact the individual, but it also affects the health of the baby, the mother–infant bond, and the marital relationship [9,10,11,12]. The entire family of the sufferer bears the burden of PPD, and the high risk of relapse can cause long-term damage [13]. Women with PPD have significantly higher medical expenses than those without PPD [14]. In fact, studies have shown that pregnant women with PPD face a higher financial burden, with an average cost of approximately USD 31,800 for mother and baby over a five-year period from pregnancy to the postnatal period [15,16].

Currently, there are two primary strategies for managing PPD. The first involves early screening in the postnatal period to identify women with PPD and provide effective treatment [17]. However, this approach is not as cost-effective as it could be because women are already experiencing PPD at the time of screening and treatment, and the cost of treating and managing PPD is prohibitive [17,18]. The second strategy involves providing prenatal psychological interventions to high-risk pregnant women to prevent the development of PPD after childbirth through early intervention [19,20,21]. However, the cost of professional psychological interventions in China is relatively high for most patients and healthcare providers, so it is unclear whether this approach is cost-effective [22].

In response to the growing concern over PPD, the Chinese government has recently implemented a series of policies aimed at exploring effective prevention and treatment options. The goal is to improve the quality of life for women suffering from PPD and reduce the harm caused by the illness [23]. Currently, these policies are in their initial stages, and their actual health economic value remains unclear. This study conducts a cost-effectiveness analysis of various PPD prevention and treatment options to determine the most cost-effective strategy in China and provide a basis for further optimization of PPD prevention and control policies.

## 2. Materials and Methods

### 2.1. Model Design

This study uses a decision tree model for the cost-effectiveness analysis (Figure 1 and Figure 2). The model used in this study is adapted from the structures of two high-quality research models [24,25]. This model has been recognized to a certain extent by the academia and has been validated in previous research. Furthermore, this study has made some modifications to the model based on the research hypotheses and the actual situation of PPD prevention and treatment in China.

By constructing a decision tree model, this study conducts a simulation study on three PPD prevention and treatment strategies. Firstly, as perinatal depression screening is not currently included in the mandatory items of maternal health check-ups in China [26], we use the strategy of not actively screening pregnant women for PPD as a routine care strategy. The second strategy is early postnatal screening. Under this strategy, doctors will actively use the Edinburgh postnatal depression scale (EPDS) to screen patients for PPD during the perinatal health check-up process and inform the patients of the screening results. Patients with positive screening results are recommended to be referred for professional psychological treatment. The last strategy is active screening plus active intervention. Under this strategy, pregnant women will receive group psychological intervention based on active early PPD screening. The primary outcome of interest in this study is the incremental cost-effectiveness ratio (ICER), which is the cost per quality-adjusted life year (QALY) gained.

Women in the early-screening group will receive routine prenatal care and basic health education during pregnancy and will undergo EPDS screening within 8 weeks postpartum. In contrast, women in the prenatal interventions group will be screened between 24 and 30 weeks of pregnancy and will receive prenatal psychological intervention if they are deemed high-risk. Both groups will undergo EPDS screening within eight weeks of delivery, and high-risk women will be referred to a specialist psychiatrist for further assessment and treatment in accordance with Chinese guidelines for managing postpartum depression. Women in the usual care group will not receive active screening and will only receive professional psychotherapy if they initiate treatment themselves.

### 2.2. Model Population

According to data from the *China Health Statistics Yearbook*, the simulated queue is assumed to consist of 10,000 pregnant women aged between 25 and 35 years old. They have no previous history of PPD and have all given birth in the past year. The time frame of this model is set to two years postpartum to reflect the majority of the impact of PPD on most affected women. The decision tree model uses Monte Carlo techniques to randomly simulate the subjects in the simulated queue [27]. In health economics, the Monte Carlo method is often used to estimate complex probability models, such as decision tree models and Markov decision models. Decision tree models often involve a large number of parameters and complex probability distributions, often making direct calculations difficult [27]. Through the Monte Carlo method, we can draw samples from the posterior distribution, and then estimate the model parameters and predict the results through these samples.

### 2.3. Criteria for Screening and Confirmation of PPD

As recommended by the Expert Consensus of the Chinese Guidelines for the Treatment of Postpartum Depressive Disorder, an EPDS score of ≥12 indicates a high risk for PPD. The diagnosis of PPD is confirmed by a structured clinical interview for DSM-IV (SCID) with a psychiatrist [28].

### 2.4. Transition Probabilities

All variables used in our decision tree models were derived from published studies (identified using PubMed and the Chinese National Knowledge Infrastructure) or authorities (China National Bureau of Statistics), as shown in Table 1. The model’s transfer probabilities adhered to a beta distribution in accordance with the recommended Chinese Pharmacoeconomic Assessment Guidelines [29].

### 2.5. Intervention and Treatment

In terms of psychotherapy and group psychological intervention programs, this study follows the recommendations in the expert consensus on perinatal depression screening and treatment published by the Chinese Medical Association. For patients with mild and moderate depression, structured psychotherapy is recommended as the first-line treatment. For patients with severe depression or those whose condition worsens after psychotherapy, a combination of psychotherapy and drug treatment is recommended as the first-line treatment. In terms of drug treatment, mainstream selective serotonin reuptake inhibitors (SSRIs) (e.g., nortriptyline, paroxetine, sertraline, imipramine, etc.) are recommended according to the guidelines. In terms of group psychological intervention, this study uses cognitive behavioral intervention as the main intervention method. The cognitive behavioral therapy program is set to conduct one hour of treatment per week for a total of six treatments [24]. The mother is screened by a psychiatrist using the EPDS and SCID to determine her risk level for PPD. Subsequently, distorted perceptions from three perspectives—self-perception, perception of the world, and perception of the future following childbirth—are gently guided and corrected.

### 2.6. Treatment Costs

In this study, treatment cost data were sourced from publicly available records provided by the Chinese Health Care Commission. The market retail price of SSRIs in China was used to determine the cost of drug treatment. Total costs were calculated by multiplying the cohort size by the sum of the costs associated with each health state [29].

### 2.7. Health Utility

The health effect values used in this study were derived from the results of the Canadian ALL OF FAMILY cohort (AOF) study [32]. Women in the AOF cohort completed a health-related quality of life (HRQoL) assessment at 4 months postpartum using a 12-item short form health survey, the 12-Item Short Form Survey (SF-12) [40]. The SF-12 comprises eight domains providing physical and mental health summary scores reflecting an individual’s HRQoL [40].

To calculate the QALYs for women, the SF-12 responses were transformed into 6-dimension SF-6D preference-based utilities, providing estimates for mothers in a depressed versus a non-depressed state [41]. Previous research has shown that the results obtained from such a conversion are credible [42]. A ‘utility’ is a weighted measure between 0 and 1 that represents individuals’ preferences for living in a specific health state [43]. Values of 1 represent perfect health and values of 0 represent a quality of life akin to being dead [43]. It should be noted that values less than 0 are also possible, where negative values represent living in a health state where quality of life is perceived to be worse than death [43]. To estimate the effectiveness of using QALYs, we multiplied women’s utilities by the time spent in each health state, as estimated from the data.

### 2.8. Model Outcome

An ICER was used as the main result of the cost-effectiveness analysis in this study. This study is based on the World Health Organization (WHO) recommendations for pharmacoeconomic evaluation [44]: ICER < 1 times gross domestic product (GDP) per capita means the increase is fully worthwhile and very cost-effective; 1 times GDP per capita < ICER < 3 times GDP per capita means the increase is acceptable and cost-effective; ICER > 3 times GDP per capita means the increase is not worthwhile and not cost-effective. China’s GDP per capita (approx. USD 11,300) is used as the willingness-to-pay (WTP) threshold.

It is assumed that screening will take place in the weeks and months after birth during follow-up appointments, as PPD does not appear immediately after birth.

### 2.9. Sensitivity and Scenario Analysis

The impact of each input parameter was assessed by a deterministic sensitivity analysis (DSA) with a one-way sensitivity analysis (±20% of the input parameter), presented as a tornado diagram [45].

Probabilistic sensitivity analysis (PSA) is used to assess the impact of parameter uncertainty on the results [45]. PSA is performed using Monte Carlo simulations with 10,000 iterations. A specific distribution was specified for each parameter in the model, with its mean equal to the point estimate.

We considered the participation of postpartum depression patients in screening and treatment, treatment pricing, and income differences among medical personnel in different regions of China. We analyzed the cost-effectiveness of prevention and treatment strategies when referral rates, hourly wages for medical personnel, and the cost of psychological screening and treatment change. Since the current income level of medical personnel in China and the cost of mental health-related services are at a low level, we only analyzed the situation where the cost of mental health services increased.

Cost-effectiveness acceptability curves (CEACs) and incremental cost and incremental quality-adjusted life year scatter plots are used to present the results of the probabilistic sensitivity analysis.

## 3. Results

### 3.1. Cost-Effectiveness Analysis

The number of episodes of PPD in 10,000 women in each PPD prevention and treatment strategy are reported in Table 2. The differences in the incidence of postnatal depressive episodes and serious complications of PPD between the three groups were highly significant. No individual cases of suicide were observed in this study.

Table 3 reports the results of the cost-effectiveness analysis. In the usual care cohort, the average total cost was USD 60.22. In the cohort that screened women for early postnatal screening after childbirth, the average total cost was USD 304.45. In contrast, the average total cost for the cohort of women who received the prenatal psychological intervention was USD 201.34. The early postnatal screening cohort and the prenatal psychological intervention cohort costs were USD 5136.85 and USD 2459.37, respectively, per QALY gained compared to the usual care cohort, which are well below the WTP of USD 11,300.

### 3.2. Sensitivity Analyses

A tornado diagram is used to illustrate the results of the one-way sensitivity analysis. The effect of extreme changes in each key parameter on the model is shown in the tornado diagram. In the model, the cost-effectiveness of screening and treatment options was not sensitive to any of the variables (Figure 3). As can be seen from the results, the cure rate associated with receiving the pharmacological/mixed treatment had the greatest impact on the outcome.

Figure 4 shows the cost-effectiveness curves for the three screening treatment strategies. The probability of being cost-effective at a WTP of USD 11,300 was 0.8%, 14.5%, and 84.7% for the usual care group, the early postnatal screening group, and the prenatal psychological intervention group, respectively. The cost-effectiveness of a prenatal psychological intervention strategy is likely to be greater than 50% when the willingness-to-pay payment is USD 400 or more per quality year (Figure 4). The cost-effectiveness of the early postnatal screening strategy will be higher than that of the usual care cohort when the WTP is greater than USD 1500.

The CEAC of the PSA results based on 10,000 Monte Carlo simulations is shown in Figure 4. The scatter is mainly in the first quadrant and the majority of the scatter is below the WTP threshold line. This suggests that the early postnatal screening strategy and the prenatal psychological intervention strategy are more cost-effective than the routine care strategy at a WTP of USD 11,300.

### 3.3. Scenario Analysis

Our scenario analysis showed that when the referral rate for women at a high risk of postpartum depression changed, the overall health benefits for the pregnant population did not show a significant increase, while the cost increase was significant (Table 3). Despite the gradual decrease in cost-effectiveness, it remained below the WTP threshold, so prenatal psychological intervention strategies remained cost-effective. When medical costs increased, the ICERs of both the early postnatal screening strategy and the prenatal psychological intervention strategy increased, but remained within an acceptable range of cost-effectiveness. This means that when the state increases subsidies for PPD treatment in pregnant and postpartum women, both medical workers and patients can receive more rewards.

**Table 3 healthcare-12-01076-t003:** The result of the base-case cost-effectiveness analysis and scenario sensitivity analyses.

	Strategy Name	COST	EFF	Incremental COST	Incremental EFF	ICER
Base-case analysis	Routine care	69.94	0.91			
	Prenatal psychological interventions	342.74	0.98	272.80	0.07	3720.74
	Early postnatal screening	451.41	0.97	381.47	0.06	6840.28
Probability of attending referral						
20%	Prenatal psychological interventions	329.76	0.98	259.81	0.07	3572.38
	Early postnatal screening	443.51	0.97	373.57	0.05	6802.90
40%	Prenatal psychological interventions	341.75	0.98	271.81	0.07	3733.13
	Early postnatal screening	454.49	0.97	384.55	0.06	6990.70
80%	Prenatal psychological interventions	355.28	0.99	285.34	0.07	3844.47
	Early postnatal screening	463.53	0.97	393.59	0.06	6904.92
100%	Prenatal psychological interventions	368.67	0.99	298.73	0.07	4032.47
	Early postnatal screening	473.83	0.97	403.88	0.06	7152.42
	Treatment costs					
Increase by 50%	Routine care	83.95	0.91			
	Prenatal psychological interventions	377.41	0.97	293.45	0.06	5229.06
	Early postnatal screening	659.39	0.96	575.45	0.05	10,938.16
Increase by 100%	Routine care	107.6835	0.91			
	Prenatal psychological interventions	486.0486	0.97	378.36	0.06	6768.28
	Early postnatal screening	845.7142	0.96	738.03	0.05	14,039.46

EFF: efficiency; ICER: incremental cost-effectiveness ratio; 20%, 40%, 80%, and 100%: referral rate level.

## 4. Discussion

This study offers a novel and systematic perspective on the exploration of this issue within the context of China. With the incidence of PPD on the rise in China, timely research is of utmost importance. Given the prolonged nature of PPD, the length of time required for treatment to yield significant improvement, the increased risk of recurrence, and the potential for negative impacts on family relationships, early screening and identification are crucial [6,46,47]. By facilitating prenatal psychological intervention, we can help women with PPD receive the timely treatment they need to mitigate the harm caused by this condition.

In the scenario analysis, we examined the impact of the referral rate for psychotherapy on the cost-effectiveness of PPD prevention and treatment strategies. The analysis results show that with the increase in the referral rate, the ICER values of both the early screening strategy group and the prenatal psychological intervention strategy group have increased. This indicates that increasing the referral rate for psychotherapy does not improve the cost-effectiveness of prevention and treatment strategies. However, from the results of the number of cases, with the increase in the referral rate for psychotherapy, the number of patients with severe depressive symptoms have significantly decreased. Although a lower referral rate can save healthcare costs to some extent, such short-sighted savings will inevitably increase the basic burden of patients, thereby causing more serious losses in the future [48]. In another scenario analysis, even when the cost of psychotherapy increases, the ICER of early-screening strategies and prenatal psychological intervention strategies still remains below one times the level of per capita GDP compared to the current standard care strategies. The increased cost of psychotherapy is largely due to the rise in income for the relevant medical staff, which may suggest that it is feasible to increase the income of psychotherapy practitioners through means such as government investment [49]. Currently, there is a huge demand for mental health services in China, yet the number of practitioners in psychotherapy are small and their general capabilities are weak [50]. Increasing investment in mental health services at the government and societal levels is expected to effectively reverse this situation [51].

Lower treatment participation rates may save public health system expenditure in the short term, but PPD can have serious long-term consequences for the health system [52]. Many women with PPD may be missed at lower treatment participation rates. Considering the adverse effects of PPD on family relationships and infant and child development, this can result in a wider impact of PPD and increased downstream health system costs. The value of screening depends on the full screening, diagnosis, and treatment pathway, which is influenced by participation and compliance rates [53,54]. Encouraging primary care physicians to become involved in PPD screening and treatment is an effective way to increase treatment participation [55]. Research has shown that women with PPD are more likely to receive contact and support from their family doctor after screening and are more resistant to visiting specialist mental health providers for treatment [56]. Establishing a comprehensive primary care maternal and child protection system to provide greater access to PPD screening and care can lead to better outcomes.

In the one-way sensitivity analysis, it was conclusively demonstrated that the model was most acutely sensitive to the cure rate of receiving medication. In the simulation, approximately 30.6% of women suffering from PPD required pharmacological treatment. However, maternity is often approached with an abundance of caution when it comes to medication. As a result, many mothers refuse to take antidepressants while breastfeeding [57], despite the fact that these drugs have been shown to pose a low risk to children who are breastfed [58,59]. Therefore, it is imperative that primary care physicians engage in health promotion efforts to educate women with PPD about safe medication options. By doing so, they can encourage adherence to treatment regimens and reduce fear and apprehension surrounding medication among women with PPD.

The value of screening for PPD remains a topic of considerable controversy and debate [60,61]. The primary point of contention centers around the role that scholars believe screening plays in reducing the incidence of PPD among pregnant women. Firstly, it is possible for a woman to have experienced adverse mental health effects during pregnancy, meaning that she may have been suffering from PPD for an extended period of time before being diagnosed postnatally [62]. Additionally, PPD may develop after postnatal screening has taken place, indicating that a single screening may not be sufficient to identify all women who experience PPD [61]. As such, early postnatal screening and prenatal psychological intervention for women at a high risk may represent a more effective approach. The results of our simulations demonstrate that both early postnatal screening strategies and prenatal psychological intervention strategies can significantly reduce the harm caused by PPD. Furthermore, prenatal psychological intervention strategies were found to be particularly effective in improving the health of women with PPD.

Regrettably, providing psychological intervention to a large number of high-risk women in China is currently hindered by three main factors. Firstly, there is a shortage of primary care physicians in China, which makes access to effective psychological counseling and health education for pregnant women a rarity [63]. Secondly, the cost of psychological interventions is often higher than the average Chinese PPD patient can afford, leading many to refuse medical treatment [64]. At present, screening and treatment for depression are only available in high-level general and psychiatric hospitals, with the primary health system bearing no relevant responsibilities. We believe that the Chinese government could introduce policies that would enable primary care facilities to provide services for mental illness.

Furthermore, there is a significant degree of discrimination in Chinese society regarding the perception of PPD. This, coupled with a lack of proper understanding of the condition among women with PPD, not only delays the optimal time for treatment but also exacerbates the patient’s condition [65]. As such, it is perhaps even more pressing to improve treatment adherence among women with PPD. At the same time, efforts must be made to correct discriminatory attitudes toward individuals with all types of mental illnesses in the Chinese society through targeted propaganda and education initiatives.

## 5. Limitations

The limitations of this study mainly stem from the variation in referral rates for patients with PPD. The referral rate parameters used in the model for this study were aggregated from studies conducted in China. China has a large population and the economic development of the country varies greatly between cities. To address this gap, sensitivity and scenario analyses were conducted for this parameter to assess how changes in referral rates would affect the results. This circumvents, to some extent, the limitations of the model in this regard. In addition, PPD does not only have adverse effects on maternal health, but can also affect infant and child development and family relationships. And these losses are difficult to estimate quantitatively, so an assessment of this was not included in this study. However, it is possible that the harm and losses caused by PPD beyond the health of the patient would make strategies for early screening and treatment of patients with PPD even more valuable.

## 6. Conclusions

In China, both early postpartum screening and prenatal psychological interventions are cost-effective strategies for preventing and treating postpartum depression. Prenatal psychological interventions for women with postpartum depression are the most cost-effective prevention and treatment strategy. Therefore, from the perspective of national payers, we recommend implementing maternal–infant screening for postpartum depression in China to identify high-risk groups early and intervene effectively. By increasing investment and expenditure in the mental health of pregnant and postpartum women, the state can not only provide better medical protection for patients but also create a better working environment for medical workers, achieving a win–win situation for both parties.

## Figures and Tables

**Figure 1 healthcare-12-01076-f001:**
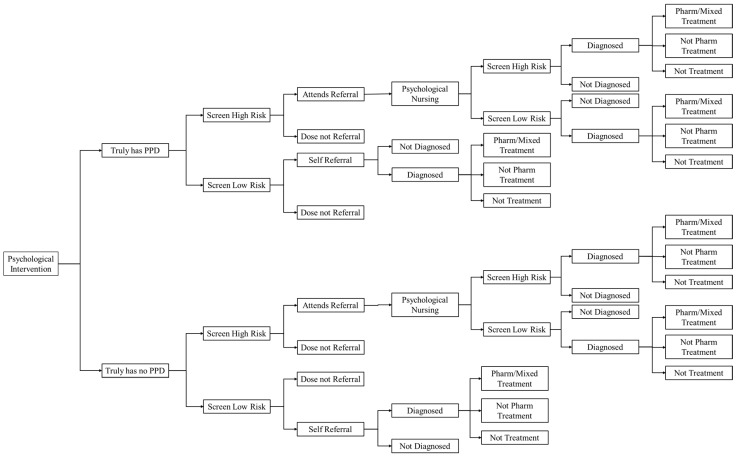
Decision analytic model (prenatal psychological intervention strategy).

**Figure 2 healthcare-12-01076-f002:**
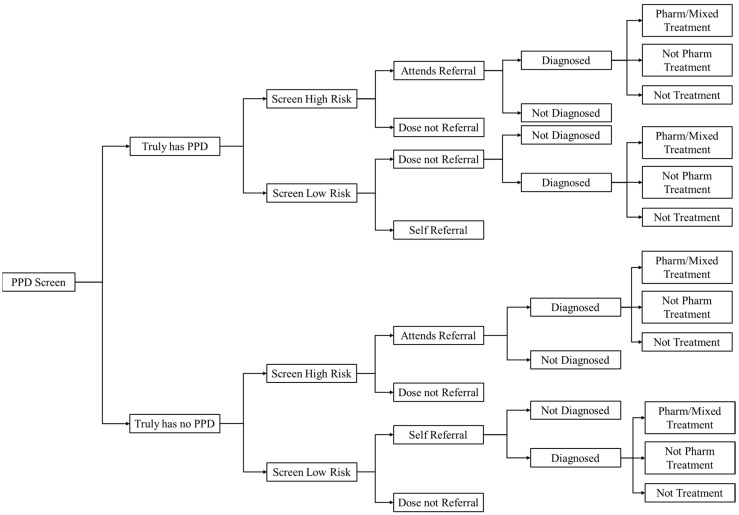
Decision analytic model (early postnatal screening strategy).

**Figure 3 healthcare-12-01076-f003:**
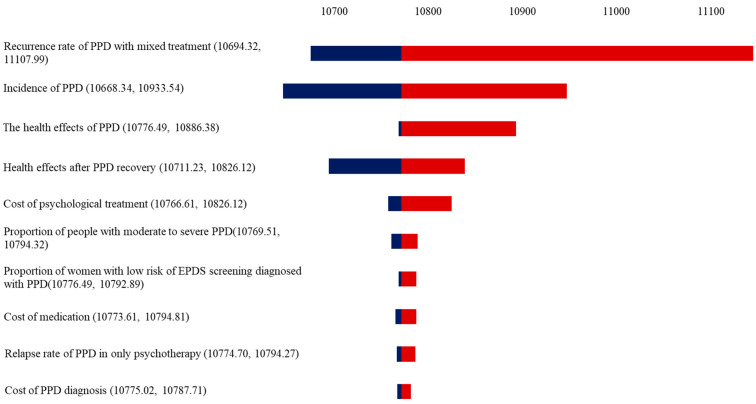
A tornado diagram showing the deterministic sensitivity analysis of the model simulation.

**Figure 4 healthcare-12-01076-f004:**
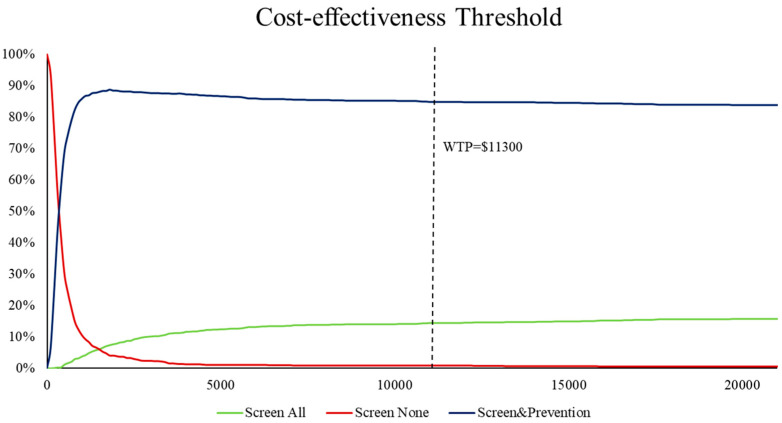
Cost-effectiveness acceptability curve.

**Table 1 healthcare-12-01076-t001:** Parameters in model.

Costs (US Dollars)				
Variables	Base Case	Range	Distribution	References
Cost of PPD care	480	576~384	Gamma	[30]
Cost of diagnosing PPD	28.57	34.284~22.856	Gamma	[31]
Cost of medication	185.75	222.9~148.6	Gamma	[31]
Cost of prenatal psychological intervention	137.15	164.58~109.72	Gamma	[30]
Cost of screening	22.3	26.76~17.84	Gamma	[30]
Cost of psychological treatment	274.3	329.16~219.44	Gamma	[30]
Health state utility				
Health	1	NA	NA	NA
Utility of having PPD	0.75	0.525~0.975	β	[32]
Utility of PPD after cure	0.88	0.616~1	β	[32]
Utility of suicidal ideation	0.38	0.266~0.494	β	[32]
Death	0	NA	NA	NA
Epidemiological parameters				
Specificity of EPDS	0.9	0.63~1	β	[33]
Sensitivity of EPDS	0.8	0.56~1	β	[33]
Probability of PPD diagnosis if high-risk	0.7	0.55~0.81	β	[32]
Probability of PPD diagnosis if low/moderate-risk	0.38	0.33~0.43	β	[32]
Probability of attending referral if high-risk	0.51	0.4~0.62	β	[34]
Probability of self-referral if low/moderate-risk	0.16	0.15~0.18	β	[34]
Probability of self-referral if unscreened	0.23	0.161~0.299	β	[34]
Probability of recovery without treatment	0.01	0.007~0.013	β	[35]
Probability of reduced risk of PPD in high-risk pregnant women after psychological intervention	0.5	0.35~0.65	β	[36]
Probability of having PPD	0.16	0.112~0.208	β	[37]
Probability of receiving pharmaceutical/mixed treatment	0.43	0.37~0.5	β	[32]
Probability of relapse from non-pharmaceutical treatment	0.2	0~0.3	β	[38]
Probability of refusal of treatment	0.17	0.15~0.19	β	[34]
Probability of women with PPD having suicidal ideation	0.117	0.0819~0.1521	β	[39]
Probability of women with PPD having suicidal behavior	0.005	0.0035~0.0065	β	[39]

PPD: postpartum depression; EPDS: Edinburgh postnatal depression scale; β: beta distribution; NA: not available.

**Table 2 healthcare-12-01076-t002:** Distribution of each health status (each group had 10,000 persons).

Referral Rate	Subject	Suicidal Ideation	PPD Recovery	PPD Unrecovered
Base-case analysis	Routine care	78	210	1336
	Prenatal psychological interventions	12	410	347
	Early postnatal screening	16	645	897
20%	Prenatal psychological interventions	31	392	401
	Early postnatal screening	32	674	944
40%	Prenatal psychological interventions	26	421	373
	Early postnatal screening	40	623	937
80%	Prenatal psychological interventions	10	478	297
	Early postnatal screening	16	640	874
100%	Prenatal psychological interventions	5	506	272
	Early postnatal screening	4	699	808

PPD: postpartum depression; 20%, 40%, 80%, and 100%: referral rate level. Subjects who underwent prenatal psychological intervention showed significant improvements in the acquisition of QALYs and in the number of postnatal depressive episodes compared to the cohort of women who underwent routine care during childbirth. The cost-effectiveness of prenatal psychological interventions was substantial.

## Data Availability

No new data were created or analyzed in this study. Data sharing is not applicable to this article.

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
