# Peer review of "An Evaluation of the Health Economics of Postnatal Depression Prevention and Treatment Strategies in China: A Cost-Effectiveness Analysis"

_healthcare, 2024, doi:10.3390/healthcare12111076_

Round 1

Reviewer 1 Report

Comments and Suggestions for Authors

Health Economics Evaluation of Postpartum Depression Treatment and Prevention Strategies in China

This study presents a compelling economic evaluation of postpartum depression (PPD) prevention and treatment strategies in China. Its primary objective is to assess the cost-effectiveness of early postnatal screening for PPD. The findings demonstrate that early postnatal screening combined with prenatal psychological interventions represents a cost-effective approach to both preventing and treating PPD.

Observations for the Authors:

The manuscript should include definitions for all abbreviations upon first use. Numerous abbreviations are employed without prior explanation.

In terms of methodology, further elaboration is needed on how the study groups were formed. Details such as whether sampling techniques were utilized or if intervention assignments were randomized should be provided.

Within the discussion section, the authors primarily focus on their own results without comparing them to findings from similar studies conducted in other countries or by other researchers investigating the cost-effectiveness of PPD prevention and treatment strategies.

The population modeling section contains a repetitive paragraph.

Reviewer 2 Report

Comments and Suggestions for Authors

Reviewer Comments

I think the research is important and interesting in terms of topic and topicality. However, there are many places in the manuscript writing that do not comply with the article writing system, and this technical error significantly reduces the quality of the manuscript. The method section of the abstract does not explain the methodology well enough. The simulation information specified in the method should also be added to the summary. In addition, the abbreviations used in the abstract should be explained with explanations since they are used for the first time. In the introduction, the justification of the study is not explained sufficiently, its difference from the studies in the literature should be revealed, it has been quoted in many places without citing the source, and the same words have been used repeatedly for the purpose of the study. There is a general spelling fluency problem in the introduction section. General information should be included at the beginning of the introduction section, then your study should be justified by revealing the studies in the literature and the difference between your research and these, and then the introduction section should be concluded by giving the purpose of the study. In this design, the introduction must be revised again. The method section is not sufficiently evidence-based. All methods used should be written clearly, citing the source. Additionally, in some places in this section, it is seen that the future tense is used as the time period. Past tense should be used when writing about a completed work. The quality of the shapes used in the results section is quite low. It is impossible to understand what is written. All figures must be edited. Explanations of abbreviations used in the tables should be added below the tables. The discussion has been made very superficially. Research results should be made much more in-depth. Additionally, most of the inferences contain personal interpretation. The discussion should be made with more sources and literature. Inferences and comments should not contain personal comments. It should be done by citing literature. Specific recommendations regarding the manuscript are as follows. Please make all suggestions carefully and completely.

Revisions

1.      Page 1, line 16; There is no explanation for the abbreviation given in the summary section. The abbreviation used for the first time must be given with an explanation.

2.      Page 1, line 10; In the method section of the Summary, add numerical data about how many people were included in the simulation and where you obtained the data. Also add information about how to perform a simulation in this section.

3.      Page 1, line 34; “PPD can strain relationships within the family unit, placing a heavy burden on all involved.” In the introduction section, the sentences mentioned should be cited by quoting from the literature. Add source.

4.      Page 2, line 44; “In fact, studies have shown that pregnant women with PPD face a higher financial burden, with an average cost of approximately $31,800 for mother and baby over a five-year period from pregnancy to postnatal[14].” Although more than one study is mentioned in this sentence, a single source is cited at the end of the sentence. If many sources are mentioned, more than one source should be added. Add the necessary resources.

5.      Page 2, line 52-54; “However, the cost of professional psychological interventions in China is relatively high for most patients and healthcare providers, so it is unclear whether this approach is cost-effective.” This sentence must be cited. If you do not cite the source, it will be understood by the readers as subjective data. Add a source related to the content of the subject to this sentence.

6.      Page 2, line 61,63; “By improving health, economic, and social outcomes for women with postpartum depression, this study will provide valuable data to support national policy formulation.” The last sentence of the introduction section is not a possible explanation. This is a sentence of expectation. However, the last sentence of the introduction should be the purpose sentence. The benefits of the study should be at the end of the discussion and in the conclusion section.

7.      Page 3, line 67; “This study uses a decision tree model for cost- effectiveness analysis (Figure 1).” Briefly explain what this model is and show the source of the model.

8.      Figures 1, 2 and 3 were made in low quality. None of the articles are read. It should be corrected by making it higher quality.

9.      Page 6, line 169; “Sensitivity and Scenario Analysis” The methods used in this section were made without citing sources. Please add the necessary sources.

10.  Page 8, line 194; “The number of episodes of PPD in 10,000 women in each PPD prevention and treatment strategy is reported in the online Supplementary Document Table S1.” Where is the online additional document table? Such a document cannot be accessed in the additional documents.

11.  Add explanations of the abbreviations you use in the table below the table.

12.  Page 11, line 257; “This groundbreaking study represents the first systematic economic evaluation of prevention and treatment strategies for PPD in China.” As stated in the sentence, avoid statements that are too assertive and describe your work as extraordinary. Make the necessary adjustments.

13.  Page 11, line 266-283; In this paragraph, information about the results is given, but in addition, many sentences are inferred without citing the source. This discussion is not appropriate to the systematic. This section is not the result section. You should make inferences by relating your results to the literature. It is not appropriate to write personal comments without citing the source. Make the necessary comprehensive edits in this paragraph.

14.   “Research has shown that women with PPD are more likely to receive contact and support from their family doctor after screening and are more resistant to visiting specialist mental health providers for treatment.” You cited studies, but you did not show the references of these studies. Add necessary resources.

Comments on the Quality of English Language

I think that the manuscript should be arranged using the English writing language and tenses in accordance with the article writing language.

Reviewer 3 Report

Comments and Suggestions for Authors

The presented research fits within the thematic scope of the journal and may contribute to the debate on preventing postpartum depression (PPD) and the associated costs of therapy. However, the article contains certain deficiencies or ambiguities that need to be addressed:

(1) Lines 46-54: The two coping strategies for postpartum depression (PPD) should be described in more detail. In other words, it is worth describing specifically the therapeutic methods within these two strategies.

(2) Line 73: Figure 1, the Decision Analytic Model, is completely illegible. Most likely, a low-resolution image was inserted. It is difficult to identify the accuracy of the proposed model.

(3) Line 221: Figure 2 (Tornado diagram showing the deterministic sensitivity analysis of the model simulation) is completely illegible. Therefore, it is difficult to assess its accuracy.

Reviewer 4 Report

Comments and Suggestions for Authors

Dear authors, I salute you for conducting the economic analysis for the preventive intervention. It is not so common, yet very important. Prior the publication, there are few minor issues that should be solved: 

Methods

Lines 94-99: This is the repetition of the paragraph above. Keep one of them.

Results

Figure 3. Please correct the image quality for this figure, the legend is almost unreadable.

Discussion

Maybe change the phrase ,,groundbreaking’’ to the ,,first study in China on this important issue’’

Line 285: Add the reference

Line 297: This is too much of the presumption

Round 2

Reviewer 2 Report

Comments and Suggestions for Authors

Reviewer Comment

1.      Page 1, line 28; Capitalize the first letters of your keywords.

2.      Page 2, line 58-63; “In response to the growing concern over PPD, the Chinese government has recently implemented a series of policies aimed at exploring effective prevention and treatment options. The goal is to improve the quality of life for women suffering from PPD and reduce the harm caused by the illness[23]. This study conducts a cost-effectiveness analysis of various PPD prevention and treatment options to determine the most cost-effective strategy in China.” At the end of the introduction section, the justification of the research and the purpose of the study are presented in a disconnected manner. Arrange the sentences in this paragraph in relation to each other.

3.      Add the abbreviations of B and NA used in Table 1 as a footnote below the table.

4.      Add the explanation of the % expression used in Tables 2 and 3 as a footnote below the table.

5.      The image quality of Figures 1, 2 and 3 is very low. A higher quality image should be added.

Comments on the Quality of English Language

It is seen that the English language of the manual is understandable, but there are places where the integrity and fluency in the transitions between sentences are not fully achieved. The minor situation needs to be corrected.

Reviewer 3 Report

Comments and Suggestions for Authors

I have no more comments.

Author Response

Thank you again for your help in improving the quality of this paper.